# Refining the resolution of the yeast genotype–phenotype map using single-cell RNA-sequencing

**Arnaud N'Guessan[1], Wen Yuan Tong[2†], Hamed Heydari[3,4], Alex N Nguyen Ba[1,2]***

[1]Department of Cell and Systems Biology, University of Toronto, Ramsay Wright Laboratories, Toronto, Canada; [2]Department of Biology, University of Toronto at Mississauga, Mississauga, Canada; [3]Department of Molecular Genetics, University of Toronto, Toronto, Canada; [4]Donnelly Centre for Cellular and Biomolecular Research, University of Toronto, Toronto, Canada

**\*For correspondence:**
alex.nguyenba@utoronto.ca

**Present address:** [†]Center of Molecular and Cellular Oncology, Yale University, New Haven, United States

**Competing interest:** The authors declare that no competing interests exist.

## eLife Assessment

This **useful** study describes expression profiling by scRNA-seq of thousands of cells of recombinant yeast genotypes from a system that models natural genetic variation. The rigorous new method presented here shows promise for improving the efficiency of genotype-to-phenotype mapping in yeast, providing **convincing** evidence for its efficacy. This article focuses on overcoming technical challenges with this approach and identifies several new biological insights that build upon the field of genotype-to-phenotype mapping, a central question of interest to geneticists and evolutionary biologists.

**Abstract** Genotype–phenotype mapping (GPM), or the association of trait variation to genetic variation, has been a long-lasting problem in biology. The existing approaches to this problem allowed researchers to partially understand within- and between-species variation as well as the emergence or evolution of phenotypes. However, traditional GPM methods typically ignore the transcriptome or have low statistical power due to challenges related to dataset scale. Thus, it is not clear to what extent selection modulates transcriptomes and whether cis- or trans-regulatory elements are more important. To overcome these challenges, we leveraged the cost efficiency and scalability of single-cell RNA sequencing (scRNA-seq) by collecting data from 18,233 yeast cells from 4489 F2 segregants derived from an F1 cross between the laboratory strain BY4741 and the vineyard strain RM11-1a. More precisely, we performed expression quantitative trait loci (eQTL) mapping with the scRNA-seq data to identify single-cell eQTL and transcriptome variation patterns associated with fitness variation inferred from the segregant bulk fitness assay. Due to the larger scale of our dataset and its multidimensionality, we could recapitulate results from decades of work in GPM from yeast bulk assays while revealing new associations between phenotypic and transcriptomic variations at a broad scale. We evaluated the strength of the association between phenotype variation and expression variation, revealed new hotspots of gene expression regulation associated with trait variation, revealed new gene functions with high expression heritability, and highlighted the larger aggregate effect of trans-regulation compared to cis-regulation. Altogether, these results suggest that integrating large-scale scRNA-seq data into GPM improves our understanding of trait variation in the context of transcriptomic regulation.

## Introduction

The process by which DNA encodes proteins via transcription and translation has been studied for decades to make sense of organisms' phenotypes. However, being able to explain organisms' phenotypes from their genetic material, that is, genotype–phenotype mapping (GPM), has been a long-lasting problem with important applications (*Bartoli and Roux, 2017*; *Ferreira et al., 2019*). Indeed, making sense of genetic variation at the phenotypic level enables the understanding of trait variation between and within species as well as the emergence and evolution of phenotypes (*Aguet et al., 2023*). For instance, reverse genetics approaches, for example, gene knockout or transgenic technologies, and forward genetics approaches like genome-wide association studies (GWAS) and quantitative trait loci (QTL) mapping helped in determining the function of multiple genes and the effects of mutations on growth in different environments (*Tarantino and Eisener-Dorman, 2012*). However, reverse genetics approaches typically fail to account for natural variation and forward genetics approaches like QTL mapping typically focus on genetic and phenotypic variation, so they cannot highlight selection on the transcriptome.

An essential characteristic of this problem is the multilayered organization of the GPM. Indeed, GPM is not strictly restricted to the direct association between genotypes and phenotypes. This association is better resolved and complemented by understanding the intermediary transcriptome layer, for example, cell mechanisms at the transcriptomic level are involved in diseases and pathogenicity (*Ferreira et al., 2019*; *Casamassimi et al., 2017*; *Wainberg et al., 2019*; *Learn Science at Scitable, 2023*; *Williams et al., 2022*). However, it is not clear to what extent transcriptomic changes relate to phenotypic changes or selection. Pioneering work from Mary-Claire King and Allan Charles Wilson set the tone for investigating this question by proposing that variations in morphological and behavioral traits arise more often through gene expression regulation than evolution at the protein-coding level (*King and Wilson, 1975*). François Jacob then postulated an essay that stemmed from this theory in which he highlights how evolution acts as a tinkerer that works from already available material, that is, through regulation of gene expression, to create new adaptations (*Jacob, 1977*). This constituted the core of the evolutionary developmental biology, which matured into the still-debated claim that new adaptations mainly emerge through cis-regulation of gene expression, that is, through noncoding DNA regulating a neighbor gene contrarily to trans-regulators acting on distant genes (*Hoekstra and Coyne, 2007*; *Kratochwil and Meyer, 2015*; *Primig et al., 2000*; *Cavalieri et al., 2000*). This debate has been reinforced by the technical difficulties and complexity of assessing the evolution and outcome of mutations in non-coding regions (*Hoekstra and Coyne, 2007*; *Kratochwil and Meyer, 2015*). For example, although human GWAS loci are mostly identified in noncoding regions and have well-characterized cis-regulatory effects, their trans-regulatory effects are usually challenging to detect (*Maurano et al., 2012*; *Schaub et al., 2012*; *Nicolae et al., 2010*). Indeed, small effect sizes, multiple testing corrections, and high false-positive rates for trans-regulation impose statistical and computational challenges for its detection (*Dutta et al., 2022*; *Wang et al., 2024*). Advances in sequencing technologies helped to partially solve these issues, particularly in the context of transcriptome analyses of the model organism *Saccharomyces cerevisiae*. For instance, *Brem et al., 2002* used microarray technology to relate the gene expression profiles of 40 yeast segregants from a lab (BY) and natural vineyard strain (RM) to their genetic markers (*Brem et al., 2002*). They found that cis-acting modulation is the main mechanism for regulating gene expression. Nearly two decades later, by greatly increasing statistical power, *Albert et al., 2018* found that most of the expression variation arises through trans-regulation using non-multiplexed RNA-seq to analyze 5720 genes in 1012 yeast segregants generated by a crossing between RM and BY (*Albert et al., 2018*). The analysis method they used, that is, expression quantitative trait loci (eQTL) mapping, consists of correlating allele frequencies to gene expression levels to find the loci modulating expression.

Although eQTL mapping is a traditional GPM analysis that accounts for the transcriptomic layer, it is typically realized through non-multiplexed RNA-seq which tends to have low statistical power due to challenges with experimental scale and confounding factors (*Schwarz et al., 2022*; *Fan et al., 2021*). Thus, eQTL mapping traditionally cannot identify significant low-effect regulatory mutations that are important for understanding the genetic bases of complex traits and diseases (*Bush and Moore, 2012*; *Lorenz and Cohen, 2012*). Furthermore, most eQTL studies only assess the average transcriptomic profile of bulk populations without being able to capture the profile of rare cell lineages within

a population. This is a critical limitation in heterogeneous populations such as cancer or microbial populations where rare lineages can drive relapse or drug resistance (*Shaffer et al., 2017*).

Here, we sought to circumvent the challenges of non-multiplexed bulk RNA-seq imposed by the scale and population heterogeneity by performing eQTL mapping through single-cell RNA sequencing (scRNA-seq) of a pool of ~4500 well-characterized F2 segregants derived from a yeast cross (*Albert et al., 2018*; *Bloom et al., 2013*; *Ehrenreich et al., 2010*). In the same way that combinatorial indexing/barcoding and multiplexing enable the collection of large-scale fitness and genotype data (*Nguyen Ba et al., 2022*), we hypothesized that scRNA-seq could help us collect both genotype and expression data on a large pool of segregants. We employ several strategies to overcome previous obstacles of eQTL mapping studies: (i) we pool cells from thousands of F2 segregants during the growth step and perform a single scRNA-seq run on the culture to account for environmental effects. The expression profiles of these lineages cannot be captured cost-effectively with bulk RNA-seq as pooling segregants in a single bulk assay would not capture individual strains' expression profiles. Moreover, (ii) from the exome sequencing data of single cells, we take advantage of the reference panel to validate that we accurately infer the genotype of each cell from extremely low number of reads mapping to polymorphic sites per cell (effectively ~0.2× coverage). Finally, (iii) we leverage the presence of thousands of high-coverage cells to train an unsupervised learning model to correct (denoise) and infer an imputed expression profile of poorly covered cells.

Using this approach, we integrated the resulting transcriptomic data from growth in rich media with a pre-existing yeast GPM to refine GPM features at a broad scale. For instance, we estimated the heritability of the transcriptome and the extent to which the transcriptome is associated with fitness. This allowed us to reveal that a negligible portion of expression variation is related to the studied phenotype variation independently of genotypic variation. This implies that almost all the phenotypic variation related to expression is also modulated by mutations. This differentiates our work from previous bulk RNA-seq studies where expression is systematically controlled for differences in lineage growth or phenotypes due to an assumption that a considerable amount of phenotypic variation is related to expression independently of genomic mutations (*Albert et al., 2018*). We also show that the increased scale of our scRNA-seq dataset enables single-cell eQTL mapping (sc-eQTL) and identifies more hotspots of expression regulation containing previously identified QTL. We also exploit the identified sc-eQTL to analyze the patterns of cis- and trans-regulation in the GPM.

## Results and discussion

### Our single-cell RNA-seq approach is consistent with yeast GPM results from non-multiplexed assays

We initially aimed to show that performing scRNA-seq at a large scale can generate data that are consistent with non-multiplexed DNA and RNA sequencing. For this purpose, we analyzed a dataset of thousands of yeast lineages generated by *Nguyen Ba et al., 2022* (*Nguyen Ba et al., 2022*). To understand the yeast GPM, they collected fitness and genotype data from ~100,000 F2 segregants derived from an F1 cross between a laboratory strain of yeast (BY) and a natural vineyard strain (RM) (*Figure 1A*).

Using this approach named barcoded bulk QTL mapping or BB-QTL mapping, they revealed the complex polygenic and pleiotropic nature of phenotypes as well as an unprecedented number of pairwise epistatic interactions. To integrate transcriptomic data into this GPM, we performed scRNA-seq using the 10X Genomics Chromium microfluidics platform (*Vermeersch et al., 2022*). Other scRNA-seq methods like Smart-seq2 offer a better breadth of coverage per cell (*Hwang et al., 2018*). However, due to the higher throughput or number of cells generated by 10X Chromium and the different coverage profiles across cells from the same F2 segregant, it is more advantageous to use 10X Chromium to capture transcriptomic variation and reconstruct the reference genomes from single cells (*Hwang et al., 2018*; *Jariani et al., 2020*). The throughput of 10X Genomics is typically in the order of $10^4$ cells which limits us to a few thousand lineages per library if we want to obtain a reasonable number of cells per lineage. This is why we focus on a single batch of 4489 F2 segregants. This method allowed us to obtain both genotype and expression profiles from 18,233 cells of the first batch of F2 segregants (*Figure 1B*). The F2 segregant barcodes are typically not expressed, so single cells have their own cell barcode, which is included in all amplified reads of the same droplet.

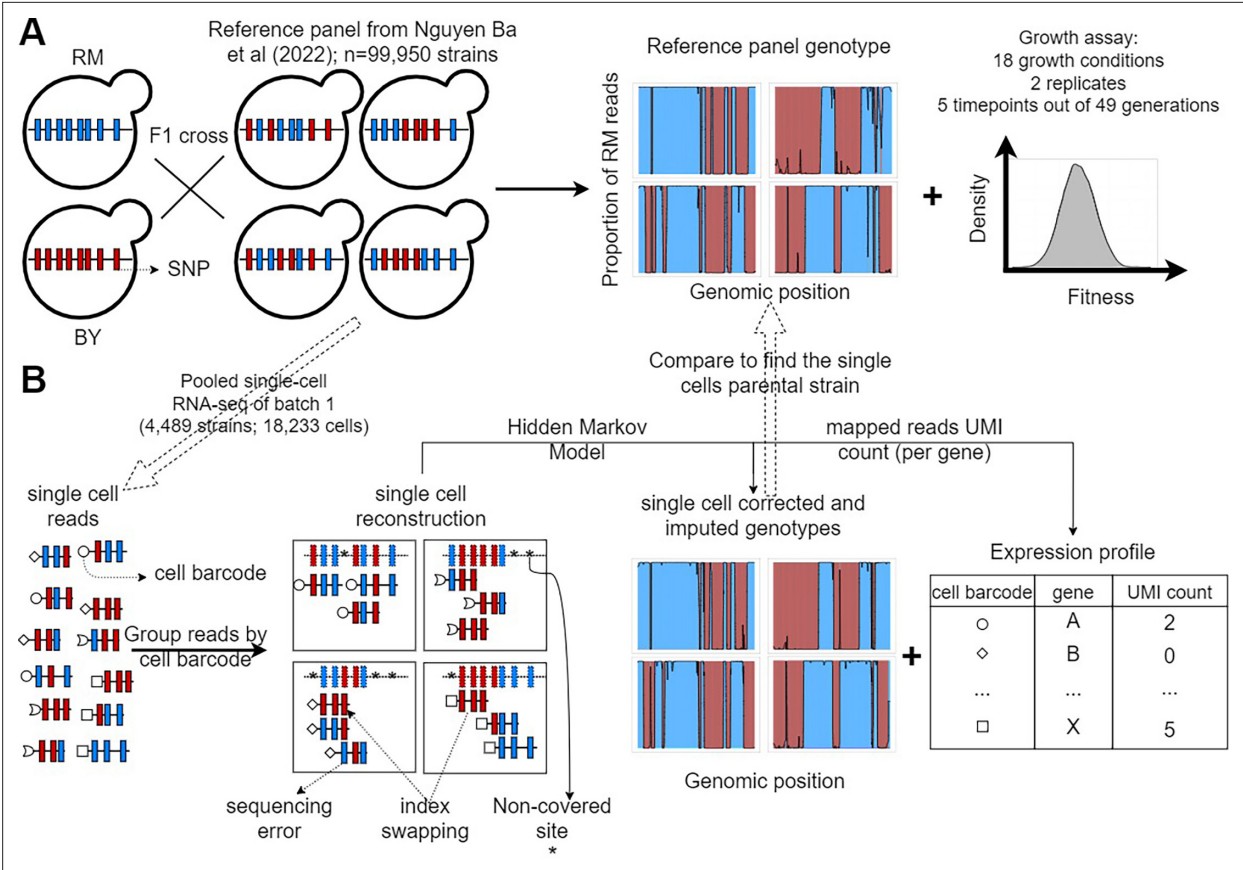

**Figure 1.** F2 yeast segregants datasets. (**A**) Reference panel and experiments from the barcoded bulk sequencing previously described in *Nguyen Ba et al., 2022*. The 99,995 F2 yeast segregants in the reference panel were derived from an F1 cross between a laboratory strain of yeast (BY) and a natural vineyard strain (RM). Thus, they only have two possible alleles at each of the 41,594 polymorphic sites. The lineage barcodes enabled fitness estimation from competition assays in 18 environments recapitulating the adaptation to temperature gradients, the ability to process different sources of carbon and the resistance to antifungal compounds. (**B**) Pooled scRNA-seq dataset from a single batch. We performed scRNA-seq of the first batch of F2 segregants (n=4489) to obtain genotypes that are similar to the reference panel and cell's expression profiles. The F2 segregant barcodes are typically not expressed, so single cells have their own cell barcode, which is included in all amplified reads of the same droplet. Non-covered sites, sequencing errors, and the presence of reads in the wrong library (index swapping) are corrected for using the Hidden Markov Models (HMM) described in *Figure 1—figure supplement 1*. Each single-cell barcode association to a reference panel strain is validated using a permutation test (*Figure 1—figure supplement 2*), and poor association was assessed using regression (*Figure 1—figure supplement 3*).

The online version of this article includes the following figure supplement(s) for figure 1:

**Figure supplement 1.** Illustration of scRNA-seq reads used to estimate the Hidden Markov Models (HMM) error rate parameters.

**Figure supplement 2.** Barcodes lineage assignment.

**Figure supplement 3.** Factors explaining the relatedness between scRNA-seq barcodes and their associated F2 segregant.

Although this short-read scRNA-seq is a high-throughput approach (*Table 1*), it comes with challenges like gene dropouts and low sequencing depth in some cells caused by technical artifacts of PCR amplification (*Lorenz and Cohen, 2012*; *Shaffer et al., 2017*).

To overcome these challenges, the unique molecular identifiers (UMIs) of the 10X Genomics platform provide a control for PCR amplification biases by quantifying gene expression from unique transcribed molecule counts instead of sequencing read counts (*Hwang et al., 2018*). However, UMIs still do not correct for dropouts or missing data in the expression profile, which could obscure potential associations between genotype and transcription (see below). We addressed this issue by fitting an imputation model called DISCERN to the expression data (*Hausmann et al., 2023*). DISCERN is a neural network that learns how to reconstruct the expression profile of high-coverage cells after embedding it in a lower dimension. This eliminates the gene dropouts and denoises the expression profile. The fact that the reconstructed expression is highly but not perfectly correlated to the original

**Table 1.** scRNA-seq dataset features.

The doublets were defined as scRNA-seq barcodes with high coverage (≥ third quartile) without significant genotype association to any F2 segregants and for which the two most similar F2 segregants do not share genotypic similarity ($R^2$ <0.1). The breadth of coverage is defined as the proportion of BY/RM polymorphic sites covered in a single cell.

| Metric | Value |
| --- | --- |
| Library size (number of reads) | 479,781,081 |
| Median number of covered genes per cell | 1,311 |
| Median number of unique molecular identifiers per cell | 5,158 |
| Doublet rate | 0.10 |
| Average number of cell barcodes per F2 segregant | 4.83 |
| Polymorphic sites mean breadth of coverage | 0.03 |

expression (mean $R^2$=50.0%, median $R^2$=52.9%; $R^2$ s.d.=10.1%) and the high variance explained by the first principal component of the imputed expression (99.6%) is consistent with an effective denoising. We take advantage of this denoised transcriptome for variance partitioning (as described later).

In addition, Hidden Markov Models (HMMs) can infer accurate genotype data even at sequencing depths as low as 0.1× (*Nguyen Ba et al., 2022*). *Nguyen Ba et al., 2022* designed an HMM to infer the segregants genotypes from the observed reads at low depth of DNA sequencing by accounting for sequencing error rate, recombination rate, and index swapping rate (*Nguyen Ba et al., 2022*). As there are only two ancestral lineages, there are only two possible alleles for the strains at each of the 41,594 polymorphic sites. Thus, the genotype of the segregants can be represented by the frequency of only one of the parental alleles, which is RM in the dataset. Applying this model to low-coverage segregants yielded genotypes that are significantly similar to high-coverage replicates (*Nguyen Ba et al., 2022*). We sought to use a similar model to infer genotypes from scRNA-seq data but we anticipated that some of these parameters may differ due to increased error rate of the reverse transcriptase, increased index swapping due to pooled reaction, etc (*Figure 1—figure supplement 1*). In Nguyen Ba et al., those rates were heuristically determined, but here we estimated these from the read mapping data and found that re-estimated parameters from data increased the proportion of recovered strains in the single cell data from 58.6% to 65.7%.

After adapting the HMM to the scRNA-seq data, we sought to validate that the resulting cell genotypes relate well to their corresponding strain in the reference panel obtained by non-multiplexed DNA sequencing strategies. Ideally, each single-cell barcode (from 10X Genomics Chromium) should be associated with a single cell and a cell should have a clear match with a unique strain in the reference panel. However, several factors can obscure these associations, for example, a single-cell droplet containing cells from two different strains, a low-coverage cell, uncertainty in the allele of the reference genotype, etc. Thus, we designed an approach to assign cells to the correct reference panel strain (see 'Materials and methods'). This approach relies on two metrics of similarity between the cells and the strains' genotypes, that is, the expected distance between them, which should be minimized for the best match, and the relatedness ($R^2$). The statistical significance of the relatedness between single cells and reference lineages was determined by a permutation test (*Figure 1—figure supplement 2*). From the read mapping alone, we obtained a mean $R^2$ of 0.59 (σ=0.19 and median = 0.64), which was significantly improved after applying our HMM to correct for misidentified alleles and imputing data in low-coverage sites using recombination probability. Indeed, the single-cell HMM genotypes yield a mean $R^2$ of 0.73 (σ=0.18 and median = 0.81; *Figure 2A*). This approach was also successful in a recent eQTL mapping on segregants with a similar genetic background (*Boocock et al., 2025*). We found that the distribution of relatedness after HMM was still left-skewed, with a minority of cells statistically significantly assigned to a reference genotype despite having what appeared to be low relatedness. Upon investigation, it was found that these could be explained by sequencing coverage and genotyping uncertainty in the single-cell and/or in the reference panel genotype(s) (*Figure 1—figure supplement 3*; *Supplementary file 1*).

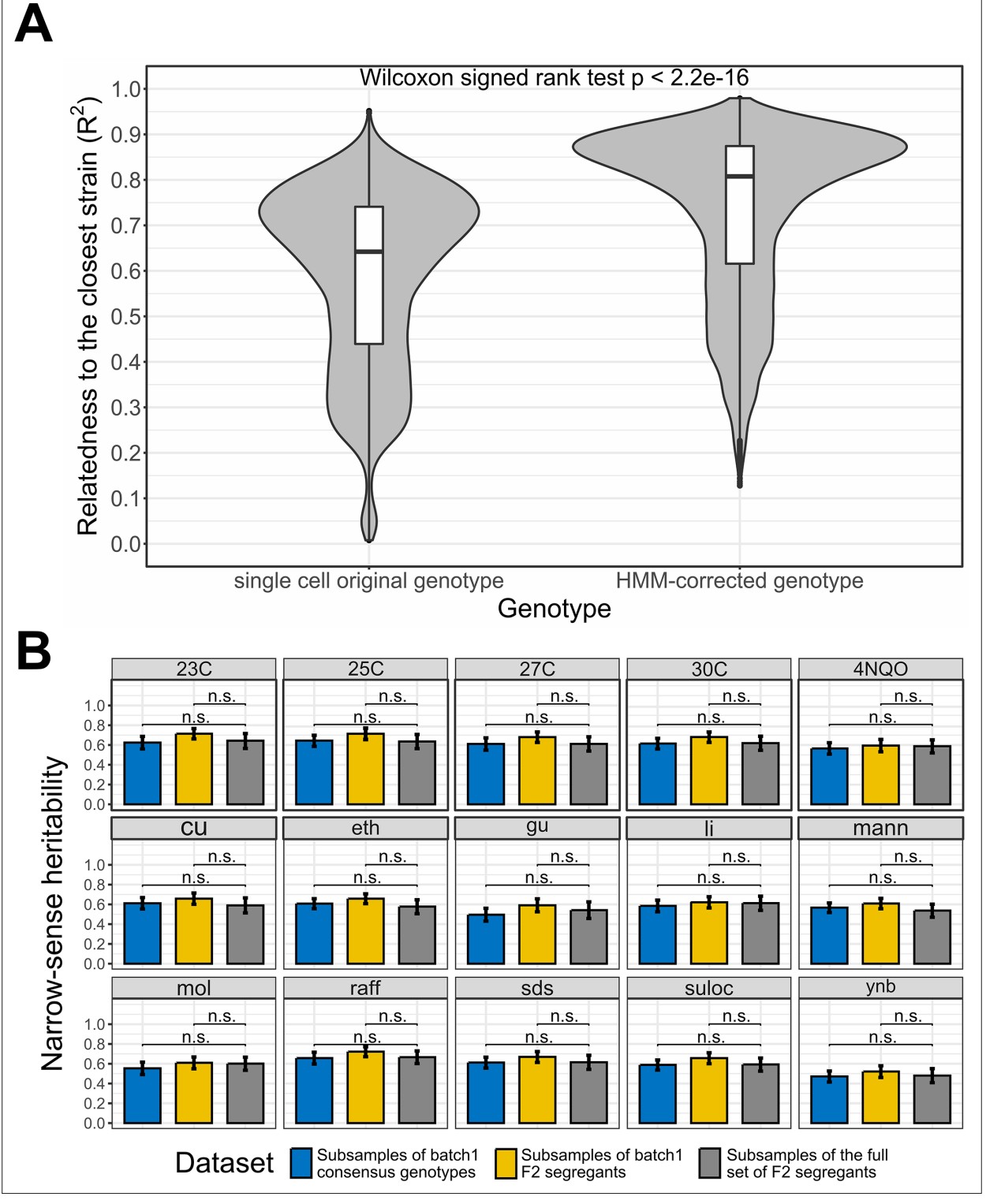

**Figure 2.** Single-cell RNA-seq data recapitulate bulk DNA and RNA assays results. (**A**) Effect of the Hidden Markov Models (HMM) on the relatedness between single-cell genotypes and their closest reference lineage. The single-cell original genotype represents the genotype of the cells before the correction with the HMM. The relatedness to the closest lineage in batch 1 has been measured with the adjusted $R^2$. To control for genotype uncertainty, only the 13,069 barcodes with a significant lineage assignment (lineage-barcode genotype correlation false discovery rate [FDR]<0.05) and a reference lineage with lower uncertainty than the single cell HMM are selected, which represents 72.2% of the barcodes. We then rounded the genotypes to remove the uncertainty during the comparison. Wilcoxon signed test p-value is indicated above the violin plots. (**B**) Narrow-sense heritability measured with scRNA-seq consensus genotypes and non-multiplexed DNA sequencing. Three datasets are compared through a *t*-test on the bootstrap

*Figure 2 continued on next page*

*Figure 2 continued*

narrow-sense heritabilities. The consensus genotypes of the batch 1 F2 segregants obtained from the associated scRNA-seq data (*Figure 2—figure supplement 1*) are represented in blue, the batch 1 F2 segregant genotypes in the reference dataset (bulk sequencing) are represented in yellow and the whole reference dataset of F2 segregants is represented in gray. For each dataset, the narrow-sense heritability has been measured from 500 random subsamples of 1000 F2 segregants. The bars represent the mean narrow-sense heritability, and the error bar length represents the 95% CI. The results are illustrated for the 15 environmental conditions where a minority of batch 1 F2 segregants have missing fitness data. The 23C-30C represents the temperature for the competition assay in YPD media while the other phenotypes represent growth on YNB, molasses (mol), mannose (Mann) or raffinose (raff) and chemical resistance to copper sulfate (Cu), ethanol (eth), guanidinium chloride (gu), lithium acetate (Li), sodium dodecyl sulfate (SDS), and suloctidil (suloc) (*Nguyen Ba et al., 2022*).

The online version of this article includes the following figure supplement(s) for figure 2:

**Figure supplement 1.** Performance of the consensus genotype methods.

**Figure supplement 2.** Comparison of the quantitative trait loci (QTL) mapping models for the 30C phenotype.

**Figure supplement 3.** Comparison of site features distribution across quantitative trait loci (QTL) models.

To further establish that the genotyping obtained from scRNA-seq data was comparable to previous non-multiplexed genotyping of the reference genotype panel, we estimated the contribution of genetic variation to the phenotypic variation, that is, fitness heritability. *Nguyen Ba et al., 2022* estimated the narrow- and broad-sense heritabilities of complex phenotypes associated with temperature gradient, carbon source, and chemical resistance for which RM and BY segregants exhibit a significant level of diversity (*Nguyen Ba et al., 2022*). We used our lineage assignment to that panel to obtain fitness but used our single-cell genotyping to perform this association ('Materials and methods'). Encouragingly, all GCTA-REML estimates of narrow-sense heritability in the scRNA-seq batch 1 dataset (blue in *Figure 2B*) are similar to the estimates from *Nguyen Ba et al., 2022* whether or not we focus on batch 1 (yellow and gray rectangles in *Figure 2B*).

Although the variance partitioning is consistent with previous studies, it only provides a broad view of the genotype-phenotype map as it does not allow the identification of the loci that significantly explain phenotype variation. If the genotypes obtained by scRNA-seq were of high quality, then we would expect that a QTL mapping model from scRNA-seq would yield a model similar to that of non-multiplexed DNA sequencing data. To achieve this, we used a cross-validated stepwise forward linear regression on the strain fitness and consensus genotypes data from single cells that shared the same lineage assignment ('Materials and methods). Performing the QTL mapping on the batch 1 scRNA-seq dataset enabled the identification of 29 QTL compared to 31 QTL identified with the bulk barcoded approach (*Supplementary files 2 and 3*; *Nguyen Ba et al., 2022*). These QTL were largely similar as shown by the non-significant difference between the effect sizes (Wilcoxon signed rank test p=0.29) and by a model similarity metric (*Nguyen Ba et al., 2022*) that considers the recombination distance between matched QTL, the similarity of the effect sizes and the allele frequencies ('Materials and methods'). Using this approach, we estimated that the similarity score between the batch 1 single cells QTL and the batch 1 BB-QTL is 86.2% while each model respectively had a similarity score of 78.7% and 78.2% with the full BB-QTL mapping performed on the 99,950 F2 segregants (*Nguyen Ba et al., 2022*; *Figure 2—figure supplement 2*). The QTL identified from the scRNA-seq dataset also recapitulated several important biological features of the reference panel such as an enrichment of non-synonymous and disordered region QTL (*Nguyen Ba et al., 2022*; *Figure 2—figure supplement 3*). Then, we compared the distributions of these two site features in the full BB-QTL dataset, which are similar in the scRNA-seq, to the distributions in the full set of SNPs using Fisher's exact test and odds ratios. We also performed the phenotype variance partitioning based on the QTL features to identify which sites contribute the most to phenotype variations. Nonsynonymous sites have significantly higher odds of being QTL than intergenic sites. The trend is modest (odds ratio = 2.17; Fisher's exact test p=1.94e-4) and the difference with synonymous sites is not significant. Furthermore, disordered sites have significantly higher odds of being QTL than intergenic sites, although the trend is modest (odds ratio = 2.00; Fisher's exact test p=1.67e-2), but the difference with structured sites is not significant.

## Integrating scRNA-seq data to an existing GPM highlights selection on the transcriptome

Having shown that our scRNA-seq allows us to relate the cells to the corresponding F2 segregant fitness data via the genotype relatedness, we next sought to highlight new associations within the BY/RM GPM. Selection is often highlighted at the genotype level through convergent evolution, an increase in allele frequency within a population, or population genetics metric (*Bergström et al., 2014*; *Good et al., 2017*; *Johnson et al., 2021*). However, the central dogma of molecular biology and the evolution tinkering model both entail that phenotype variation should be linked to transcriptomic variation. As our dataset included all these variables, we sought to evaluate the expression-genotype association strength and provide a variance partitioning framework to evaluate the association between the transcriptome and trait variation ('Materials and methods'). The GCTA-REML variance partitioning model used to estimate trait heritability in the previous section can also be modified to include gene expression as the response variable and cell genotypes as the only random effect ('Materials and methods'). This enables the quantification of expression heritability, that is, the variance of expression explained by genotype. Using this approach on our large-scale dataset, we estimated that genotype explains 75.7% of expression variance. This estimate is weighted for expression levels as genes that are more expressed account for more of the expression variation. At the individual gene level, the genotype explains, on average, 35.4% of transcript level variation ($\sigma=4.51\%$; min = 31.0%; median = 33.4%; max = 100%), which is slightly but significantly higher than previous estimates from non-multiplexed eQTL mapping studies (Wilcoxon signed rank p<2.2e-16). Indeed, *Albert et al., 2018* estimated that at the gene level, the genotype explains on average 30.0% of the expression variance using a lower-power dataset of 5720 genes in 1012 yeast segregants generated by the same parental strains (RM and BY).

In addition to expression heritability, our 'one-pot' approach and the multidimensionality of our scRNA-seq dataset enable the evaluation of the association between traits and expression. This was previously challenging with large-scale bulk RNA-seq experiments as they require batching strategies. This results in batch effects, which are linked to environmental and technical variations that affect lineages' growth rates and expression. To eliminate this bias, *Albert et al., 2018* controlled gene expression levels for variation in growth rates across lineages, making it impossible to correlate fitness to gene expression. Measuring the gene expression profiles of thousands of lineages simultaneously removes the need for batching strategies as it allows us to collect enough transcriptomes to understand the associations between different components of the genotype-phenotype map, which we demonstrate with the 30C phenotype (*Figure 3*).

The components of this variance partitioning all relate to at least one biological phenomenon. Indeed, the portion of trait variation explained exclusively by the genotype variation (red in *Figure 3*) represents the effect of mutations on fitness via several biological phenomena, such as protein stability, enzymatic function, etc, independent of expression level. For the 30C phenotype, this component explains 20.9% of the fitness variation in the BY/RM background, which is lower than the 47.8% explained by the shared component between phenotype, genotype, and expression variations (purple in *Figure 3*). The latter represents the association between selection (fitness) and the transcriptome either through loci influencing fitness via expression directly, for example, a mutation in the promoter, or through loci affecting expression via an effect on cell fitness (indirectly) (*Sun et al., 2020*; *Marguerat and Bähler, 2012*). Its considerable association with fitness variation thus supports the evolution and tinkering model for the 30C phenotype. As for the phenotype variation explained exclusively by gene expression (blue in *Figure 3*), it could represent epigenetics and stochastic gene expression, which weakly explain variations in the 30C phenotype (0.69%). Moreover, the total variance explained by the model is consistent across sample sizes, highlighting the robustness of the GREML method. However, increasing the sample size allowed an increase in the relative contribution of the genotype–expression interaction component at the expense of the other components, suggesting that increasing the scale of the dataset increases the confidence of the associations between expression and genotype changes that contribute to trait variation. We also note that denoising the expression data with DISCERN slightly increases the contribution of the genotype-expression interaction and decreases the residual of the model by up to 8% (*Figure 3—figure supplement 1*).

Although this model explains a large proportion of the 30C phenotype variance and its heritability, the residuals still represent ~30% of fitness variation (*Figure 3*). This could be explained by

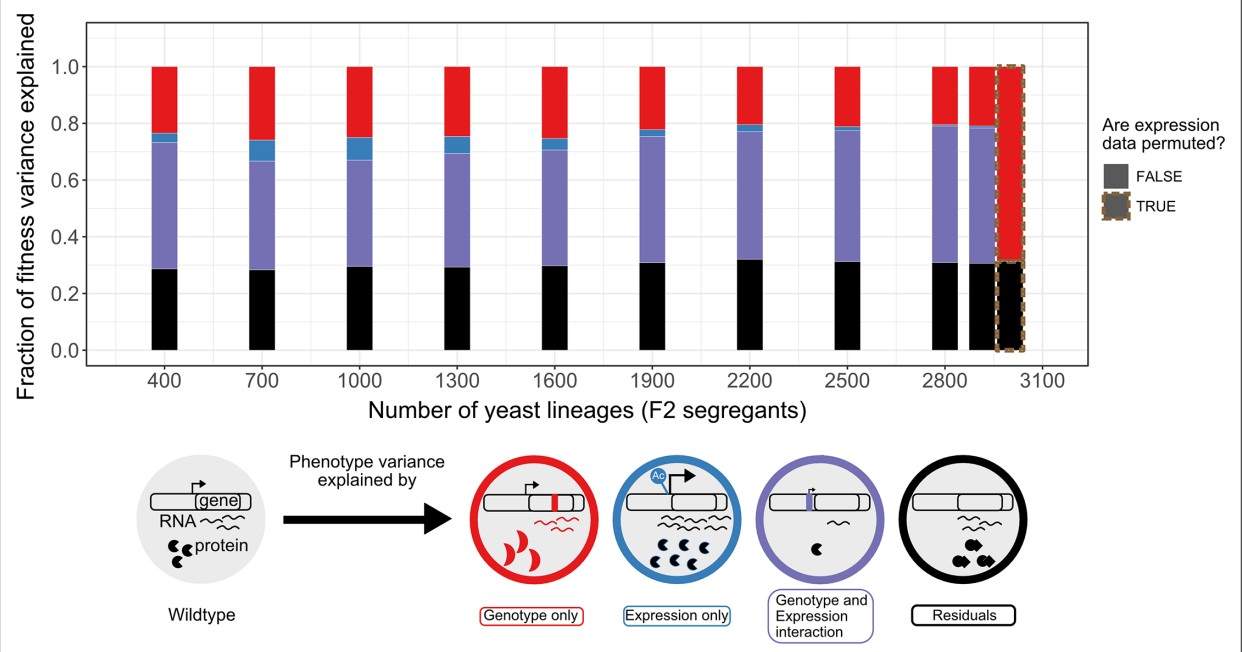

**Figure 3.** Variance partitioning of the 30C phenotype from scRNA-seq. We fitted F2 segregant fitness data to their cells' consensus genotype and expression profiles using GTCA-GREML at different sample sizes. Because each segregant had multiple representatives/cells in the scRNA-seq data, the consensus profile of these lineages has been obtained from the cell with the highest transcript level. The bars represent the proportion of the fitness/phenotype variance across F2 segregants (whole rectangle area) explained by each model component. The size of the red genotype-exclusive component is obtained by subtracting the size of the shared component from the variance explained by the model in *Equation 2*. The red cell illustrates this effect as a mutation changes its fitness by modifying the structure of the transcript product. The size of the blue expression-exclusive component is obtained by subtracting the size of the shared component from the variance explained by the model in *Equation 3*. An epigenetic modification increasing the transcript level of the gene in the blue cell illustrates this effect. The purple area represents the effect of the interaction of mutation and expression on fitness variation observed across the segregants. In the example shown, a mutation in the promoter changes the fitness of the purple cell through a down-regulation of the gene expression. This component is obtained by subtracting the variance explained by the model in *Equations 1* or 2 from the one in *Equation 4*, as there is a shared component of trait variation explained by genotype and expression variation. Finally, the black bar represents the model's residual, which includes any phenomenon that causes fitness variation that we did not explicitly measure and include in the model. The size of the residuals is obtained by subtracting the sum of the size of all the defined components (red + blue + purple) from 1.

The online version of this article includes the following figure supplement(s) for figure 3:

**Figure supplement 1.** Denoising the expression data with DISCERN decreases the residuals of the variance partitioning model.

unmeasured factors like high-order epistasis. However, the broad-sense heritability of this phenotype is similar to the narrow-sense heritability, suggesting that the residuals are mostly not explained by genotype and expression (*Nguyen Ba et al., 2022*). *Nguyen Ba et al., 2022* also estimated that epistasis only explained around 5% of fitness (*Nguyen Ba et al., 2022*). Other factors like mitochondrial mutations, post-transcriptional changes, protein–protein interactions, or other protein properties could play a role in explaining the residual of the model and are good candidates to extend this analysis. These results suggest that a single run of scRNA-seq on a single batch of F2 yeast segregants converges with bulk DNA sequencing results while refining broad-scale features of the GPM.

## The increased scale of scRNA-seq enables the discovery of new eQTL

Our integrative scRNA-seq approach is not limited to enabling the quantification of the association between transcriptomic changes and trait variation. Indeed, the same approach we used to identify QTL can be used to detect loci regulating gene expression which can reveal the cell mechanisms underlying trait variation through transcriptomic changes. We thus modified the QTL mapping framework such that the response variable is the level of expression of a single gene in the single cells ('Materials and methods'). Because our cells are not synchronized, differences in the cell cycle could create transcriptomic variation that could obscure the association with genotypic variation. To control for this, we averaged out the effect of cell cycle by reconstructing the consensus expression and

genotype profiles of each segregant significantly associated to the cells. This approach is a cost-efficient way to perform eQTL mapping from the expression profile and genotype of cells from thousands of lineages in a multiplexed way (sc-eQTL mapping).

Consistent with yeast non-multiplexed eQTL results, the genes with the highest expression heritability are enriched in functions related to carbohydrate catabolic process (GO:0016052) and cellular biosynthetic process (translation GO:0006412, organelle assembly GO:0070925, ribosome biogenesis GO:0042254 and gene expression GO:0010467) (Fisher's exact test false discovery rate FDR<0.05; 'Materials and methods'). In both datasets, these genes are also highly expressed, which reflects the positive correlation between expression heritability and expression levels ($R^2$=0.81 and p<2.2e-16). Although some of these highly expressed genes have important functions modulated by mutations, their higher heritability could also be explained by higher statistical power to measure heritability when expression counts are higher. Moreover, genes with the lowest expression heritability observed in the RM/BY background, which we defined as the bottom 10% expression heritability, are enriched in functions related to the cell cycle biological process (GO:0007049, Fisher's exact test FDR <0.05) which is consistent with bulk RNA-seq eQTL (*Albert et al., 2018*; *Mi et al., 2019*). This does not suggest that genes with this function are not important for variation in gene expression at the whole-genome scale but rather that mutations observed in the RM/BY background are not correlated with variation in the expression of these specific genes. This could reflect conservation for genes involved in the cell cycle biological process (*Bähler, 2005*; *Yu et al., 2006*).

Because of the increased scale of our collection, our approach is more powered to estimate the gene expression heritability. Indeed, we detected a median of 21 eQTL per gene (μ=34.3, σ=29.7), which is almost four times higher than what has been detected by the largest non-multiplexed RNA-seq dataset in the same genetic background (*Albert et al., 2018*). This is expected as the power to detect trait loci is positively correlated to the sample size (*Nguyen Ba et al., 2022*). We validated this trend in our dataset by performing a power analysis with the 10 most heritable genes (*Figure 4—figure supplement 1*; linear regression adjusted $R^2$=0.95 and p<0.05), which had the following functions: glycolysis, gluconeogenesis, pentose phosphate pathways, AMP metabolism, and chaperone. These newly detected eQTL probably have small effect sizes as they only allow to increase the expression heritability by 5% on average. However, for the 3730 out of 6240 genes (59.8%) that have higher expression heritability in our dataset, this increase is on average 15.7%. We were also able to detect new overrepresented biological processes, that is, the DNA metabolic process (GO:0006259) and the response to nutrient levels (GO:0031667), for which the variation of expression levels is weakly associated with the genetic variation observed across the F2 RM/BY segregants.

The functional enrichment analysis using scRNA-seq data revealed new associations between expression heritability and biological processes in the RM/BY genetic background. However, while it suggests that many eQTL are also QTL, it cannot accurately point to the specific loci involved in trait variation and cannot address whether mutations on regulatory hubs have stronger effects on traits. To investigate this, we mapped the QTL to hotspots of gene regulation (or regulatory hubs), which we defined as 25 kb genomic windows that were repeatedly identified in the eQTL mapping procedure (for different genes). This was done to acknowledge the uncertainty in the exact position of the eQTL due to linkage disequilibrium and power. We then ranked the 30C QTL identified by *Nguyen Ba et al., 2022* based on their absolute effect size and correlated it to the rank of the eQTL hotspots based on the number of regulated genes. This resulted in a positive correlation (Spearman $\rho$ =0.33 and p=5.21e-5), suggesting that larger effects on the regulatory network translate into larger trait variation. Indeed, we observed that some previously reported high-effect-size QTL genes are located in eQTL hotspots, for example, MKT1, HAP1, and IRA2 (*Figure 4A*).

Performing this rank test on individual genes also yielded the result that eQTL effect is correlated with the fitness effect for 35.1% of the genes (permutation test p<0.05, see 'Materials and methods'). Although this correlation does not apply to most genes, it highlights potential regulatory mechanisms explaining the importance of the strongest growth loci or QTL. For instance, MKT1, that is, the strongest growth loci, is part of a regulation hotspot affecting genes that are important for yeast growth like ENP1 which is involved in RNA processing and HXT6 which is involved in glucose uptake (*Roos et al., 1997*; *Chen et al., 2003*; *Roy et al., 2015*). Among the strongest growth loci, VPS70 is part of a hotspot of regulation that strongly affects the expression of RSF2, a zinc-finger protein regulating glycerol-based growth and respiration (*Lu et al., 2005*). Furthermore, the highest peaks for expression

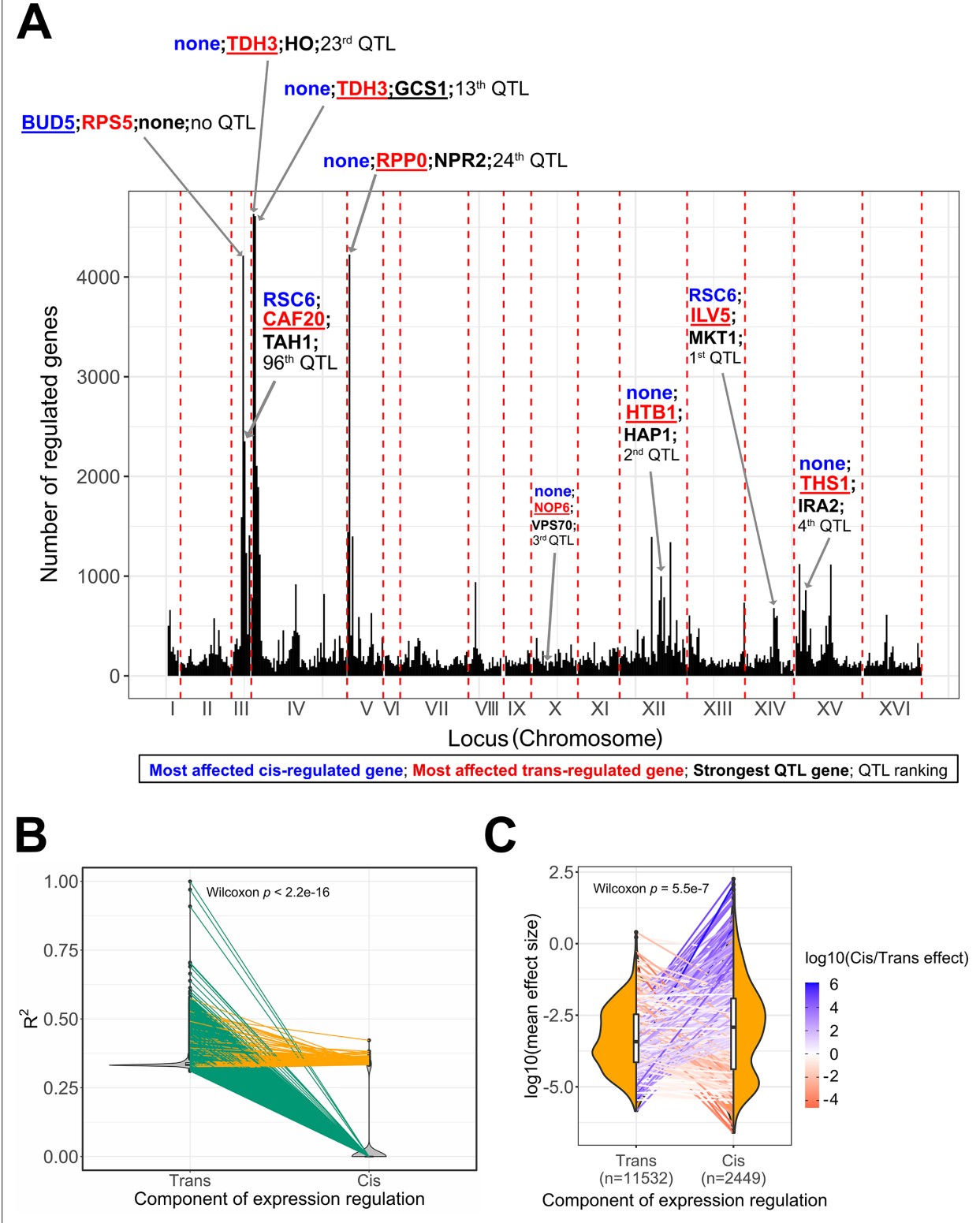

**Figure 4.** Expression quantitative trait loci (eQTL) features underlying trait variation across the BY/RM segregants. (**A**) Mapping of the 30 C QTL in the eQTL hotspots. We represent the hotspots of expression regulation as genomic windows (25 kb) to acknowledge the uncertainty around the real position of the eQTL due to linkage disequilibrium. We annotated the top 5 eQTL hotspots and the eQTL hotspots in which the top additive QTL identified by the BB-QTL mapping of the 30C phenotype are located. In these regions, we represented the most affected trans-regulated genes in red, the most affected cis-regulated gene in blue, and the top QTL genes in black. The double quotation characters represent the absence of such

*Figure 4 continued on next page*

*Figure 4 continued*

genes in the associated region. We also represented the rank of the QTL in the set of 159 QTL of the 30C phenotype. (**B**) Partitioning of the expression heritability or explained variance ($R^2$) among cis- and trans-eQTL. Each pair of points connected by a line represents a gene. Green lines represent the genes that only have trans-eQT,L and orange lines represent the genes that have both trans- and cis-eQTL. (**C**) Comparison of the mean effect size between cis- and trans-eQTL. Each pair of points connected by a line represents a gene. The line color represents the ratio of the average effect size between cis- and trans-eQTL. The sample size of each eQTL category is represented in the x-axis. This is the number of trans-eQTL and cis-eQTL used for calculating the average effect sizes per gene not the number of points per distribution.

The online version of this article includes the following figure supplement(s) for figure 4:

**Figure supplement 1.** The number of detected expression quantitative trait loci (eQTL) revealed by scRNA-seq increases with sample size.

regulation contain important growth loci on chromosome IV (13th and 23rd strongest QTL among 159 QTL; *Figure 4A*). This hotspot is also responsible for regulating TDH3, which is involved in glycolysis and gluconeogenesis and can have an important effect on fitness (*Vande Zande et al., 2022*). A recent eQTL mapping study on segregants with a similar genetic background that also leverages one-pot scRNA-seq and genotype inference also identified such hotspots of regulation and highlighted their role in the trade-off between reproduction and cell cycle progression (*Boocock et al., 2025*).

The presence of these hotspots suggests that expression differences in BY/RM would predominantly be explained by mutations in trans-regulatory elements. To test this, we partitioned the variation in gene expression between cis- and trans-regulatory loci for each gene (see 'Materials and methods'). This analysis revealed that all the genes are affected by at least one polymorphic trans-regulatory locus and that these polymorphic trans-regulatory loci explain most of that gene's expression (*Figure 4B*). It is well known that mutations in promoters and nearby enhancers can influence gene expression (*Mattioli et al., 2020*; *Romero et al., 2012*). Indeed, we identified many genes that contained an allele in a cis-regulatory element that strongly explains that gene's expression variation (n=750 genes out of 6088, *Figure 4B*). As expected, mutations in cis-regulatory elements were of stronger effect size than trans-eQTL individually, but the cumulative aggregate effect of all trans-eQTL acting on that gene was comparable to the few cis-eQTL they had (*Figure 4C*). This can be explained by the fact that there are more opportunities for mutations to arise in trans-regulatory elements. Finally, we found that trans-eQTLs have two times higher odds of affecting cell fitness than cis-eQTL ($\chi^2$ p=0.01). Furthermore, by repeating our eQTL hotspot analysis with *Albert et al., 2018* data, we observed a non-significant association between eQTL hotspot and QTL ($\chi^2$ p=0.50). In that bulk RNA-seq dataset, QTL have less odds of being in eQTL hotspots compared to other loci (odds ratio = 0.72) while in this scRNA-seq dataset, QTL odds of being in eQTL hotspots are two times higher than other loci. This could be explained by the smaller sample size of that bulk RNA-seq dataset and by the fact that they control for differences in growth across lineages. Taken together, these results suggest that the link between the genetic basis of transcription variation across RM/BY segregants and fitness could only be revealed by integrating large-scale transcriptomic data to an existing GPM, which scRNA-seq facilitates.

## Conclusion

By leveraging the scalability of scRNA-seq, we obtained thousands of transcriptomes from a reference pool of strains in a single experiment. This enabled the analysis of the association between genotype, transcriptome, and phenotype at an unprecedented scale. Questions surrounding transcriptomic variation and phenotypic variation have been at the center of many previous quantitative genetics studies (*Wang et al., 2024*; *Brem et al., 2002*; *Albert et al., 2018*; *Bloom et al., 2013*; *Brem et al., 2005*; *Bloom et al., 2019*; *Westra et al., 2013*; *Yao et al., 2017*; *Liu et al., 2019*). For instance, how strong is the connection between the transcriptome and trait variation? Although most GWAS loci are located at noncoding regions and are well-characterized for cis-regulation effects, is trans-regulation more important for phenotypic and expression variation? What are the gene functions that are the most heritable and modulable at the expression level? These ideas and discoveries all support the fact that researchers can gain valuable insight into the evolution of traits by integrating the transcriptome in GPM analyses, which can translate into fundamental knowledge or other important applications where phenotypes evolve.

In this study, we took advantage of a previously characterized BY/RM cross where the genetic basis of growth in various environments was examined in detail (*Nguyen Ba et al., 2022*). By integrating

transcriptomic data in this genotype-phenotype map, we identified transcriptome components that are involved in trait variation and evaluated their contribution. Similar to a previous study, which obtained transcriptomes by individual strain sequencing, we found that gene expression is highly heritable. Further, our study design also allowed us to conclude that gene expression contributes to a significant portion of the phenotypic variation in this strain collection.

This finding is corroborated by our findings that most eQTL detected in our study were previously shown to be QTL. This is perhaps not surprising given that QTL in this cross were previously inferred to be in regulatory genes, but this provides a more mechanistic view of the effect of an allele on phenotype. Indeed, we find a bias for trans-regulation for generating transcription innovation where the cumulative effect of trans-eQTL on gene expression is significant. That is not to say that cis-regulatory alleles are dispensable as cis-regulatory alleles often have a large effect on gene expression. This is similar to studies on human transcriptomes where trans-eQTL are cumulatively more impactful and enriched in sets of complex trait loci (*Wang et al., 2024*; *Westra et al., 2013*; *Yao et al., 2017*; *Liu et al., 2019*). This genome-wide view of the genetic basis of transcriptional variation has consequences for the evolution of phenotypes as the target size afforded by trans-eQTL is far larger than cis-eQTL. Thus, adaptation to small and fluctuating environmental changes may proceed preferentially through allelic changes or recombination of many small-effect trans-eQTL, but large expression changes are likely to require some cis-eQTL.

In this study, we leveraged the availability of genotype and phenotype data on our pool of strains. This was obtained by liquid handling robotics and pooled competitive growth assay with barcode sequencing. While this was performed on a very large scale, it was essentially obtained by brute force and through approaches that are not necessarily applicable to other systems. Although it is clear from our results that single-cell sequencing can achieve the same genotype quality as single-reaction genotyping, it is much harder to obtain phenotype data from scRNA-seq. Thus, our framework might not be readily translatable to other systems where similar studies on the GPM are desirable. However, two observations from this cross can be used to suggest an experimental approach. First, while epistasis is important, it contributes to a relatively small portion of the phenotypic variance. Second, transcriptomic variation contributes little to the missing heritability. Thus, it may be possible to use predicted fitness instead of observed fitness and recapitulate findings from this study. Indeed, the phenotype under study has a negligible component of its variance explained by transcription independent of the genotype, but this feature was unknown before the study. This alternative approach can only relate expression to the heritable component of traits, which would have missed this variance explained by transcription independent of the genotype and explains why we did not employ it. Predicted fitness could be obtained from bulk-segregant analysis where the additive effect of loci can be inferred from whole-genome sequencing (*Ehrenreich et al., 2010*; *Brauer et al., 2006*). However, it is unclear whether the proposed experimental approach is generalizable in other contexts, and collecting fitness data may not be necessary if a GPM analysis is already available for the studied system.

Despite the study's limitation on generalizability, our scRNA-seq framework helps bridge the understanding of how genetic variation influences transcriptomic variation at a broad scale. Our framework relies on identifying the genome of single cells from the transcriptome, which is going to be possible from low-coverage sequencing when genetic variation within the pool is high (such as this cross, microbiome sequencing, or cancer cells with extensive copy number variation), and from low cell diversity with sufficient transcriptomic variation such that aggregation of single cells with similar transcriptomes can afford pseudo-high coverage sequencing. Although we restricted the scope of this work to broad-scale transcriptomic and genomic patterns related to trait variation, we demonstrated that it is possible to integrate large-scale transcriptome data into an existing GPM, which can be exploited in the future for more fine-scale analysis like co-expression analysis and highlighting how specific trans-regulatory patterns within the cell network lead to trait variation. For instance, identifying new eQTL could enable a better understanding of the gene network by highlighting the downstream trans-regulatory effects of mutations and their association with trait variation. For most of the genes, this newly discovered eQTL explained more than 15% of their transcript level variation on average and could help characterize how cells react to environmental changes. Moreover, the fact that it is an integrative approach allowed us to evaluate the association between phenotypes, expression, and mutations. Although we did not validate the strength of this association for multiple traits and biological systems, this approach is readily available to help screen for traits that are evolving

independently of eQTL if weak correlations between phenotypic variation and genotype–expression interaction are observed, that is, phenotypes driven by epigenetics. Finally, the residuals of the model could help hypothesize on the contribution of unmeasured phenomena like protein–protein interaction, gene content variation, or post-transcriptional modification, although it could be modest. Thus, integrating genotype, transcriptome, and phenotype using scRNA-seq data can be particularly efficient for developing a better understanding of other important traits or diseases.

## Materials and methods

### Yeast strains and segregants

We analyzed cells from a single batch (batch 1) of 4489 F2 segregants obtained from a F1 cross between the yeast laboratory strain BY4741 and the vineyard strain RM11-1a generated in a previous study (*Nguyen Ba et al., 2022*). These strains have been selected to generate this collection of segregants because they exhibit differences in multiple phenotypes including the adaptation to temperature, the ability to process different sources of carbon, and the ability to resist antifungal compounds. Therefore, the genetic variation observed across the segregants can be correlated to the differences in growth rate observed in the 18 environments recapitulating these phenotypes in the *Nguyen Ba et al., 2022* study (*Nguyen Ba et al., 2022*). The selection of the batch is random, and the fact that we performed the analyses on a single batch eliminates batch effects that could obscure variable associations. Genotypes and fitness data used were the same ones obtained in the previous study.

### Yeast growth and single-cell RNA-sequencing protocol

To prepare strains for scRNA sequencing, we unfroze the batch of segregants and inoculated approximately $5*10^6$ cells in YPD (1% yeast extract, 2% peptone, 2% dextrose) to saturation. The next day, about $10^7$ cells were passaged to 5 mL of fresh YPD and grew for 4 hours to bring cells to log phase. We then pelleted 100 ul of cells and resuspended them in spheroplasting solution (5 mg/mL zymolyase 20T, 10 mM DTT, 1 M sorbitol, 100 mM sodium phosphate pH 7.4) at a concentration of $10^7$ cells/mL. The cells were incubated at 37°C for approximately 10 minutes at which point spheroplasting was verified by mixing a small aliquot of cells with detergent to observe lysis. The cells at this point were quantified using a hemocytometer and prepared using the standard 10X Genomics Gel Beads-in-emulsion (GEM) protocol. We used the Chromium Next GEM Single-cell 3' Reagent Kit to prepare the sequencing libraries and sequenced on a NextSeq 500 high-output flow cell.

We note that the cells analyzed here were grown in bulk and assayed for their transcriptome in log phase. Our fitness data was obtained from competitive bulk fitness assays which include several whole growth cycles over multiple days and thus capture lag phase, exponential growth, and saturation. Nevertheless, previous experiments had shown that fitness was mostly determined by exponential growth, which suggests that our analysis is adequate even if the cells were prepared for sequencing at a single time point.

### Single-cell RNA-sequencing data parsing

From the scRNA-seq reads, we obtained gene expression levels and allele counts using the pipeline count from CellRanger version 3.1.0 (*Zheng et al., 2017*). For each of the ancestral strains, that is, RM11-1a and BY4741, the pipeline mapped the scRNA-seq reads to the reference genome, filtered the barcodes by comparing the UMI count per barcode distribution to a background model of empty gel-bead in-emulsion, and counted the number of UMI per gene per barcode. The barcode filtering retained 18,233 barcodes. For each barcode, we then counted the number of RM and BY alleles at each polymorphic site by parsing the RM and BY bam files using a Python script (https://github.com/arnaud00013/sc-eQTL/tree/main/II_scRNA-seq_genotyping). This script only keeps reads that mapped at the same loci on both reference genomes to increase the level of confidence of the mapping.

### Correction and imputation of single-cell genotypes with an HMM

Because there are only two possible alleles at each polymorphic site of the F2 RM/BY segregants, their genotype can be recapitulated by a quantitative variable measuring the proportion of reads from one of the parental strains, which is RM in our dataset. The raw allele count data provides a

first estimate of this RM allele frequency at each polymorphic site. However, due to the low mean depth of coverage of scRNA-seq data (0.2×), the absence of reads in some polymorphic sites and the biases introduced during sequencing like index hopping/swapping, we expect that the raw data can be imputed and corrected for errors and uncertainty in the observed alleles. Therefore, we applied a Hidden Markov Model (HMM) on the observed allele count. Such a model can infer accurate genotype data at sequencing depths as low as 0.1× (*Nguyen Ba et al., 2022*; *Hwang et al., 2018*; *Stark et al., 2019*). *Nguyen Ba et al., 2022* designed an HMM to infer the segregants genotypes from bulk DNA sequencing by accounting for sequencing error rate, recombination rate, and index swapping rate (*Nguyen Ba et al., 2022*). Because scRNA-seq uses the reverse transcriptase, which has a higher error rate, and because it is a pooled assay with higher chances of index swapping, we expected the HMM parameter to differ for the single cell data. Therefore, we adapted the HMM to scRNA-seq data by measuring its parameters in our dataset (*Figure 1—figure supplement 1*). The scripts are available on GitHub (https://github.com/arnaud00013/sc-eQTL/tree/main/II_scRNA-seq_genotyping).

## Assigning single cells to the reference panel strains

To evaluate the level of relatedness between the reference panel strains and the imputed single-cell genotypes, we used the expected distance to identify the strain that best relates to each single cell:

$$Expected\ distance\ (g_c, g_s) = \sum_{i=1}^{41594} g_c + g_s - 2g_cg_s \tag{1}$$

where $g_c$ is the cell genotype and $g_s$ is the strain genotype. Next, we assigned the single cell to its best match in the studied batch of 4489 trains only if this match is better than the best match in randomly generated batches of the same size (*Figure 1—figure supplement 2*). This procedure is implemented and available at https://github.com/arnaud00013/sc-eQTL/tree/main/III_Genotype_analysis.

## Partitioning the phenotypic variance into genetic and transcriptomic components

To analyze the yeast GPM at a broad scale and evaluate the association between selection and the transcriptome, we estimated the contribution of genetic and transcriptomic variations to phenotypic variation from scRNA-seq data. More precisely, we performed a Genome-wide Complex Trait Analysis (GCTA) by fitting a linear mixed model to the data using the restricted maximum-likelihood (REML) method (*Yang et al., 2011*):

$$y = X\beta + W_gu_g + \varepsilon_g \tag{2}$$

$$y = X\beta + W_eu_e + \varepsilon_e \tag{3}$$

$$y = X\beta + W_gu_g + W_eu_e + \varepsilon \tag{4}$$

where $y$ is the fitness vector for the $n$ segregants with a significant association to batch 1 cells, $X$ is the $n \times k$ matrix of $k$ fixed effects, $\beta$ is the vector of $k$ coefficients of the fixed effects, $W_g$ is the $n \times p$ genotype matrix, $u_g$ is the vector of $p$ SNP effects, $W_e$ is the $n \times m$ expression matrix, $u_e$ is the vector of $m$ gene expression effects, and $\varepsilon$ is the error term. To decrease the effect of gene dropouts and noise in the gene expression profile of each cell, we fitted an imputation model called DISCERN to the expression data (*Hausmann et al., 2023*). DISCERN is an autoencoder that learns how to reconstruct the expression profile of high-coverage cells after embedding it in a lower dimension. These operations eliminate the gene dropouts and denoises the expression profile. We also controlled for the effect of the cell cycle on gene expression by using the consensus F2 segregants expression and genotypes from the associated cell data. The consensus data has been obtained using the cell with the maximum read count. Because the dataset does not include fixed effects, we set the fixed effect to a vector of ones such that its coefficients represent the mean fitness while the genotype and expression data are the random effects that explain the fitness variance along with the error terms. The REML solution assumes that the data follow a Gaussian distribution, so the data are standardized before fitting the model. We also divided the standardized expression counts by the cell sum of expression counts to control for molecule count biases across cells. The cell fitness is based on the fitness of the closest segregant in batch 1 as measured by the expected distance. Because this model is linear and additive, it can be compared to the estimates of narrow-sense heritability obtained by *Nguyen Ba et al.,*

*2022* (*Nguyen Ba et al., 2022*). The difference between the variance explained in *Equation 4* and *Equation 2* or 3 allow to infer the variance explained only by the genotype or the expression component of the model. The code for the variance partitioning is available on GitHub ([https://github.com/arnaud00013/sc-eQTL/tree/main/IV_variance_partitioning](https://github.com/arnaud00013/sc-eQTL/tree/main/IV_variance_partitioning)).

### Estimating the expression heritability from scRNA-seq

To obtain this estimate from scRNA-seq data, we considered the fact that GCTA-REML only takes a vector as a response variable while the gene expression matrix is multi-dimensional. To solve this, we orthogonalized the gene expression matrix using principal component analysis (PCA) and used its first PC, which explains 99.6% of the variance, as a response variable of the model. In case 99% of the expression matrix is explained by more than one PC, it is possible to use a weighted sum to obtain the expression heritability. Indeed, if the expression PCs recapitulate the total expression variance and are orthogonal or independent to each other, then the sum of the PCs variance explained by genotype should be the expression heritability. If '$k$' is the number of PCs explaining at least 99% of expression variance:

$$Expression\ heritability = \sum_{i=1}^{k} PC_i\ eigen\ value\ *\ \text{``}PC_i \sim genotype\text{''}\ \ model\ R^2 \tag{5}$$

### QTL mapping

To identify the loci that influence cell fitness, we performed a linear regression on the consensus genotypes of the strains from the scRNA-seq data and the strain fitness. We decided to use the consensus genotypes of the strains as they relate better to the bulk segregant genomes. To build the consensus genotypes, we defined cells from the same lineage as the ones that shared the same closest F2 segregant in batch 1. Next, we used the median to obtain cells' consensus genotypes as it is less sensitive to outliers and because it yields the best relatedness to the batch 1 reference genotypes (median $R^2$=87.0%; μ=79.5%; σ=18.2; *Figure 2—figure supplement 1*). We selected the QTL in the linear models using cross-validation on the scRNA-seq data. This analysis consists of dividing the dataset into 10 random partitions of similar sample sizes and running a cross-validated stepwise forward linear regression on each partition. For each partition, the model starts with no QTL and a linear model 'Fitness ~Genotype' is fitted using the genotype data at each polymorphic site, where the correlation coefficient represents the effect size of the SNP. Then, the forward search starts and at each iteration, a new locus with the minimum linear model residual sum of squares (RSS) is added to the QTL model, which is updated with new effect sizes after the addition of a new SNP. Because the order of addition of QTL matters in the forward search and because some QTL are linked or collinear, the model can be refined by exploring different QTL around the local optima. These steps are repeated until the model RSS cannot be improved anymore or until the number of QTL reaches an arbitrary maximum far from the cross-validated number of QTL. After the forward search is completed in each partition, the algorithm calculates the optimal $\lambda$ values that minimize the objective function $F_o$:

$$F_o\left(\beta\right) = RSS\left(\beta\right) + Lasso\ penalty\left(\beta\right)$$

$$\|Y - X\beta\|_2^2 + \lambda\,\|\beta\|_0 \tag{6}$$

where $\beta$ is the vector of SNP effect sizes in the QTL model, $\|Y - X\beta\|_2^2$ is the RSS of the linear QTL model, $\lambda$ defines the penalty for adding a new SNP to the model, and $\|\beta\|_0$ is the number of SNPs in the QTL model. This objective function has the property to add sparsity in the QTL model and thus avoid overestimating the number of QTL while being consistent (*Nguyen Ba et al., 2022*). The optimal $\lambda$ has a minimum of log(n) which corresponds to the Bayesian information criterion (BIC), which is known to yield correct models asymptotically (*Harrell, 2001*). This allows us to consider the possibility that a sparser model than the one found using the BIC could yield better predictive power on a test set while avoiding overfitting. The optimal $\lambda$ values found in all the partitions are then averaged and the resulting mean $\lambda$ is used to solve the objective function in the full dataset, which yields the optimal QTL model. The cross-validation assumes that the partitions are independent, such that the variance explained by the model and the number of relevant QTL are unbiased estimates.

## Highlighting hotspots of gene regulation through eQTL mapping

To identify the loci regulating gene expression regulation, we adapted the QTL mapping framework using expression as the predicted phenotype. Because this approach had to be repeated for each of the 6240 genes, we needed to modify it so that the execution time was convenient. To do so, the parameter $\lambda$ was not estimated using cross-validation but rather from the BIC, that is, $\lambda = \log(n)$ where n is the number of cells. We found that the BIC was often selected by the cross-validation procedure when tested on a few genes and thus we do not believe that this approach will significantly change our results.

To acknowledge the uncertainty around the exact position of eQTL due to linkage disequilibrium, we define eQTL hotspots as 25 kb genomic windows that were repeatedly identified in the eQTL mapping procedure. Finally, we defined hotspots of gene regulation as genomic windows in the fourth quantile of the distribution of the number of regulated genes per window and the fourth quantile of the number of eQTL per window. The code for the single-cell eQTL mapping is available on GitHub (https://github.com/arnaud00013/sc-eQTL/tree/main/V_sc_eQTL_mapping).

## Functional enrichment analysis by gene ontology annotation

To highlight gene functions enriched at different levels of expression or expression heritability, we performed the Panther database binomial test for statistical overrepresentation of gene ontology biological processes (*Mi et al., 2019*; *Thomas et al., 2022*). A low level was defined as within the 25% bottom part of the distribution (<Q1) while a high level was defined as within the top 25% part of the distribution (>Q3). The p-values were corrected for multiple testing using the false discovery rate correction (FDR).

## Matching QTL to eQTL

To evaluate the contribution of gene expression regulation to fitness variation, we created a model to match QTL and eQTL based on the similarity of loci and the similarity of predicted effect on gene expression. More precisely, for each of the 6088 genes for which we could detect eQTL, we performed a new eQTL model by correlating the expression level of the gene to the genetic variation at QTL positions. This allowed us to measure the predicted effect of the QTL on gene expression. We then calculated the distance between the QTL and the real eQTL of the gene based on recombination distance within each chromosome, which decreases exponentially with genetic distance, and the difference in the predicted effect on the gene expression using the formulation developed by *Nguyen Ba et al., 2022*. Next, we used the same Needleman–Wunsch algorithm to find the most likely set of pairing between QTL and eQTL, where an unmatched QTL is also possible but penalized. Finally, we determined the proportion of genes for which gene expression regulation is associated with higher fitness. To do so, for each gene, we performed a permutation test by comparing the average rank of the matched QTL of the gene to the average rank of 999 random subsets of unmatched QTL of the same size. The p-value is the proportion of random subsets of unmatched QTL with a higher average QTL rank than the set of matched QTL.

## Comparing cis- and trans-eQTL contribution to expression variation

We used the definition of local eQTL in *Albert et al., 2018* to define cis-eQTL, that is, any eQTL between 1000 bp upstream of the gene and 200 bp downstream of the gene. Thus, we defined trans-eQTL as the eQTL that do not follow this criterion. For each gene, we then performed variance partitioning using the GCTA:

$$y = X\beta + W_{g\_cis}u_{g\_cis} + \varepsilon_{\_cis} \tag{7}$$

$$y = X\beta + W_{g\_trans}u_{g\_trans} + \varepsilon_{trans} \tag{8}$$

$$y = X\beta + W_{g\_cis}u_{g\_cis} + W_{g\_trans}u_{g\_trans} + \varepsilon \tag{9}$$

where $y$ is the vector of expression level of the gene across the $n$ cells, $X$ is the $n \times k$ matrix of $k$ fixed effects, $\beta$ is the vector of $k$ coefficients of the fixed effects, $W_{g\_cis}$ is the $n \times p$ cis-eQTL genotype matrix, $u_{g\_cis}$ is the vector of $p$ cis-eQTL effects on expression, $W_{g\_trans}$ is the $n \times m$ trans-eQTL expression matrix, $u_e$ is the vector of $m$ trans-eQTL effects on expression, and $\varepsilon$ represent the error terms. Because the dataset does not include fixed effects, we set the fixed effect to a vector of ones such

that its coefficients represent the mean expression level while the cis-eQTL and trans-eQTL genotypes are the random effects that explain the expression variance along with the error terms. We can infer the variance explained by the cis-eQTL by the difference in variance explained between the models in *Equations 8 and 9*. Likewise, the difference of variance explained by the models in *Equations 7 and 9* can help us estimate the variance explained by the trans-eQTL. Finally, we estimate the effect sizes using the absolute value of the correlation coefficients of each loci and compare the median between the cis- and trans-eQTL from the same gene (paired data) with a Wilcoxon signed rank test.

## Acknowledgements

ANNB acknowledges support from the Natural Science and Engineering Research Council of Canada (NSERC RGPIN-2021-02716 and DGECR-2021-00117), and AN acknowledges support from the Natural Science and Engineering Research Council (NSERC CGS-D App Id: 569340-2022, UTF Fellowship from the University of Toronto and Rustom H Dasture Graduate Scholarship in Cell and Systems Biology). The computations in this paper were enabled by resources provided by Compute Canada. We also thank the Centre for the Analysis of Genome Evolution and Function (CAGEF) and TCAG at SickKids for sequencing support.

## Additional information

### Funding

| Funder | Grant reference number | Author |
|---|---|---|
| Natural Sciences and Engineering Research Council of Canada | RGPIN-2021-02716 | Alex N Nguyen Ba |
| Natural Sciences and Engineering Research Council of Canada | DGECR-2021-00117 | Alex N Nguyen Ba |
| Natural Sciences and Engineering Research Council of Canada | CGS-D: 569340-2022 | Arnaud N'Guessan |

The funders had no role in study design, data collection and interpretation, or the decision to submit the work for publication.

### Author contributions

Arnaud N'Guessan, Conceptualization, Data curation, Formal analysis, Investigation, Methodology, Writing – original draft, Writing – review and editing; Wen Yuan Tong, Conceptualization, Investigation; Hamed Heydari, Software; Alex N Nguyen Ba, Conceptualization, Data curation, Formal analysis, Supervision, Funding acquisition, Validation, Investigation, Methodology, Writing – original draft, Project administration, Writing – review and editing

### Author ORCIDs

Arnaud N'Guessan  https://orcid.org/0000-0002-3385-725X
Alex N Nguyen Ba  https://orcid.org/0000-0003-1357-6386

Reviewer #1 (Public review): https://doi.org/10.7554/eLife.93906.5.sa1
Reviewer #2 (Public review): https://doi.org/10.7554/eLife.93906.5.sa2
Author response https://doi.org/10.7554/eLife.93906.5.sa3

## Additional files

### Supplementary files

Supplementary file 1. Predictors of the relatedness between high-coverage cells and their closest batch 1 lineage. We used these variables as explanatory variables in a linear regression model where

the relatedness of the high-coverage cells was the response variable. The high-coverage cells were defined as having a coverage in the top 25% of the distribution (>Q3). The data were normalized before performing the linear regression and the influence of the predictors is ranked in decreasing order from top to bottom of the table.

Supplementary file 2. QTL identified from the bulk fitness and DNA sequencing assays.

Supplementary file 3. QTL identified from single cells HMM-corrected genotypes and closest lineage fitness.

MDAR checklist

## Data availability

The code used for this study is available and explained at https://github.com/arnaud00013/sc-eQTL (copy archived at **N'Guessan, 2025**) and the original single-cell reads from the pooled segregants scRNA-seq assay have been uploaded in the NCBI BioProject database with the accession number PRJNA1022775. The single-cell barcodes expression data are also available at https://github.com/arnaud00013/sc-eQTL as an archive file named Matrix_gene_expression_barcodes_1_to_9000.csv.tar.gz or Matrix_gene_expression_barcodes_9001_to_18233.csv.tar.gz.

The following dataset was generated:

| Author(s) | Year | Dataset title | Dataset URL | Database and Identifier |
|---|---|---|---|---|
| N'Guessan A, Tong WY, Heydari H, Nguyen Ba AN | 2025 | Refining the resolution of the yeast genotype-phenotype map using single-cell RNA-sequencing | https://www.ncbi.nlm.nih.gov/bioproject/?term=PRJNA1022775 | NCBI BioProject, PRJNA1022775 |

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
