## [Editor Report · eLife Assessment]

This **useful** study describes expression profiling by scRNA-seq of thousands of cells of recombinant yeast genotypes from a system that models natural genetic variation. The rigorous new method presented here shows promise for improving the efficiency of genotype-to-phenotype mapping in yeast, providing **convincing** evidence for its efficacy. This article focuses on overcoming technical challenges with this approach and identifies several new biological insights that build upon the field of genotype-to-phenotype mapping, a central question of interest to geneticists and evolutionary biologists.

---

## [Referee Report · Reviewer #1 (Public review)]

N'Guessan et al have improved the report of their study of expression QTL (eQTL) mapping in yeast using single cells. The authors make use of advances in single cell RNAseq (scRNAseq) in yeast to increase the efficiency with which this type of analysis can be undertaken. Building on prior research led by the senior author that entailed genotyping and fitness profiling of almost 100,000 cells derived from a cross between two yeast strains (BY and RM) they performed scRNAseq on a subset of ~5% (n = 4,489) individual cells. To address the sparsity of genotype data in the expression profiling they used a Hidden Markov Model (HMM) to infer genotypes and then identify the most likely known lineage genotype from the original dataset. To address the relationship between variance in fitness and gene expression the authors partition the variance to investigate the sources of variation. They then perform eQTL mapping and study the relationship between eQTL and fitness QTL identified in the earlier study.

This paper seeks to address the question of how quantitative trait variation and expression variation are related. scRNAseq represents an appealing approach to eQTL mapping as it is possible to simultaneously genotype individual cells and measure expression in the same cell. As eQTL mapping requires large sample sizes to identify statistical relationships, the use of scRNAseq is likely to dramatically increase the statistical power of such studies. However, there are several technical challenges associated with scRNAseq and the authors' study is focused on addressing those challenges. The authors have successfully demonstrated their stated goal of developing, and illustrating the benefit of, a one-pot scRNA-seq experiment and analysis for eQTL mapping.

---

## [Referee Report · Reviewer #2 (Public review)]

This work describes the single-cell expression profiling of thousands of cells of recombinant genotypes from a model natural-variation system, a cross between two divergent yeast strains.

I appreciate the addition of lines 282-291, which now makes the authors' point about one advantage of the single-cell technique for eQTL mapping clearly: the authors don't need to normalize for culture-to-culture variation the way standard bulk methods do (e.g. in Albert et al., 2018 for the current yeast cross), and without this normalization, they can integrate analyses of expression with those of estimates of growth behaviors from the abundance of a genotype in the pool. The main question the manuscript addresses with the latter, in Figure 3, is how much variation in growth appears to have nothing to do with expression, for which the answer the authors given is 30%. I agree that this represents a novel finding. The caveats are (1) the particular point will perhaps only be interesting to a small slice of the eQTL research community; (2) the authors provide no statistical controls/error estimate or independent validation of the variance partitioning analysis in Figure 3, and (3) the authors don't seem to use the single-cell growth/fitness estimates for anything else, as Figure 4 uses loci mapped to growth from a previously published, standard culture-by-culture approach.

---

## [Author Response]

The following is the authors’ response to the previous reviews

**Reviewer #2:**
Minor reviews:The caveats are (1) the particular point will perhaps only be interesting to a small slice of the eQTL research community; (2) the authors provide no statistical controls/error estimate or independent validation of the variance partitioning analysis in Figure 3, and (3) the authors don't seem to use the single-cell growth/fitness estimates for anything else, as Figure 4 uses loci mapped to growth from a previously published, standard culture-by-culture approach. It would be appropriate for the manuscript to mention these caveats.

We have added two small mention of these caveats – mainly that the study may not generalize, and that the study does not attempt to try the variance partitioning on other traits or other system where the values of the partitions are better established.

I also think it is not appropriate for the manuscript to avoid a comparison between the current work and Boocock et al., which reports single-cell eQTL mapping in the same yeast system. I recommend a citation and statement of the similarities and differences between the papers.

We have added this reference and a clear statement of similarities between the two studies. It was not our intention to avoid this; we had simply not seen that study in the initial submission.